# Insights into Gradient and Anisotropic Pore Structures of Capiox^®^ Gas Exchange Membranes for ECMO: Theoretically Verifying SARS-CoV-2 Permeability

**DOI:** 10.3390/membranes12030314

**Published:** 2022-03-10

**Authors:** Makoto Fukuda, Ryo Tanaka, Kazunori Sadano, Asako Tokumine, Tomohiro Mori, Hitoshi Saomoto, Kiyotaka Sakai

**Affiliations:** 1Department of Biomedical Engineering, Kindai University, 930 Nishimitani, Kinokawa-City 649-6493, Japan; 1718360051s@waka.kindai.ac.jp (R.T.); 1718360065r@waka.kindai.ac.jp (K.S.); tokumine@waka.kindai.ac.jp (A.T.); 2Industrial Technology Center of Wakayama Prefecture, 60 Ogura, Wakayama-City 649-6261, Japan; tomohiro_mori@wakayama-kg.jp (T.M.); saomoto@wakayama-kg.jp (H.S.); 3Department of Applied Chemistry, School of Advance Science and Engineering, Waseda University, 3-4-1 Okubo, Tokyo 169-8555, Japan; kisakai@waseda.jp

**Keywords:** extracorporeal membrane oxygenator (ECMO), polymethylpentene (PMP), polypropylene (PP), SARS-CoV-2, water vapor permeation

## Abstract

When using the extracorporeal capillary membrane oxygenator (sample A) for ECMO treatments of COVID-19 severely ill patients, which is dominantly used in Japan and worldwide, there is a concern about the risk of SARS-CoV-2 scattering from the gas outlet port of the membrane oxygenator. Terumo has launched two types of membranes (sample A and sample B), both of which are produced by the microphase separation processes using polymethylpentene (PMP) and polypropylene (PP), respectively. However, the pore structures of these membranes and the SARS-CoV-2 permeability through the membrane wall have not been clarified. In this study, we analyzed the pore structures of these gas exchange membranes using our previous approach and verified the SARS-CoV-2 permeation through the membrane wall. Both have the unique gradient and anisotropic pore structure which gradually become denser from the inside to the outside of the membrane wall, and the inner and outer surfaces of the membrane have completely different pore structures. The pore structure of sample A is also completely different from the other membrane made by the melt-extruded stretch process. From this, the pore structure of the ECMO membrane is controlled by designing various membrane-forming processes using the appropriate materials. In sample A, water vapor permeates through the coating layer on the outer surface, but no pores that allow SARS-CoV-2 to penetrate are observed. Therefore, it is unlikely that SARS-CoV-2 permeates through the membrane wall and scatter from sample A, raising the possibility of secondary ECMO infection. These results provide new insights into the evolution of a next-generation ECMO membrane.

## 1. Introduction

Hollow fiber membrane oxygenator is used for extracorporeal membrane oxygenation (ECMO) [1,2,3,4] to treat acute respiratory distress syndrome caused by coronavirus disease (COVID-19). Extracorporeal membrane oxygenator (ECMO) is the “last stronghold” for patients with COVID-19 severe coronavirus infection [1]. Thanks to the devoted efforts of medical professionals, such as Japan ECMOnet, the weaning rate of COVID-19 critically ill patients in Japan is 67%, which is one of the highest in the world [3].

Dysfunctions of ECMO, such as the excessive pressure drop due to blood coagulation in the blood flow path [5], condensation of water vapor (wet lung) and plasma leakage [6,7,8], and the decrease in the gas exchange rate due to the decrease in effective membrane area have been reported. Of these, it was indicated that the condensation of water vapor within the pores of the hollow fiber membrane was unlikely to be the cause of plasma leakage [6].

Vapor condensation and plasma leakage are more likely to occur in ECMO support, which takes a longer time than cardiovascular surgery, such as open heart surgery. For this reason, in Japan, apart from the “extracorporeal membrane oxygenator for open heart surgery”, the “ECMO for assisting circulation/ECMO for assisting respiration” for ECMO support has been approved [9]. Both are expected to be used within a period of 6 h. The requirement for the “ECMO for assisting circulation/ECMO for assisting respiration” is that “the membrane characteristics of the silicone membrane or the special polyolefin membrane can prevent plasma leakage”. Plasma leakage is less likely to occur with silicone or special polyolefin (polymethylpentene) membrane [7].

However, it has been reported that plasma in the blood permeated into the lumen of the hollow fiber membrane through the membrane wall and leaked as a yellow foamy liquid from the gas outlet port of the ECMO in the actual ECMO supports of COVID-19 critically ill patients [2]. At that time, there was a concern that SARS-CoV-2 in plasma might permeate into the lumen of the hollow fiber membrane and scatter as the aerosol from the gas outlet port [2]. If the pore diameter of the hollow fiber membrane is 50 nm or more, there is a risk that SARS-CoV-2 with a diameter of 50–200 nm [10] permeates through the membrane wall. In this case, a SARS-CoV-2 positive reaction was detected even when plasma leakage was not visible [4].

In this regard, there is a concern that SARS-CoV-2 may leak from Capiox^®^ (“ECMO for assisting circulation/ECMO for assisting respiration”, Terumo Co., Ltd., Tokyo, Japan) which is dominantly used in Japan and worldwide [11,12,13,14]. Terumo Co., Ltd. has two types of Capiox^®^ on the market, “extracorporeal membrane oxygenator for open heart surgery” and “ECMO for assisting circulation/ECMO for assisting respiration”, which have been approved. Two types of Capiox^®^ membranes are formed by the microphase separation processes (thermally induced phase separation (TIPS)) [12] using paraffin as a solvent, and other membranes are formed by melt-extruded stretch processes [15,16,17,18]. 

The objective of this study is to clarify the membrane structure of two types of Capiox^®^ gas exchange membranes and verify the risk of SARS-CoV-2 leakage using the approach of our previous study [19,20,21,22,23,24,25,26,27]. We focus on the durability impaired by water vapor permeation through the membrane wall and condensation in the inner lumen of the hollow fiber membrane. In addition, the relation between the long-term durability and the membrane structure formed by the microphase separation or the stretching method is noteworthy for developing the next-generation ECMO membrane in comparison with the previous studies.

## 2. Material and Methods

### 2.1. Hollow Fiber Membranes for Extracorporeal Membrane Oxygenator

The technical data of the samples are shown in Table 1. Capiox^®^ CX-LX2 LW (Terumo Co., Ltd., Tokyo, Japan, Sample A) has built-in hollow fiber membranes made from poly (4-methyl-2-pentene) (PMP) [28,29]. BIOCUBE^®^ (NIPRO Co., Ltd., Tokyo, Japan) is shown for comparison [19]. A module design and technical data were shown in our previous study [19,30] to examine the blood flowing outside the hollow fiber. These are the outside blood flow oxygenators in which blood perfuse the outside of the hollow fiber membrane.

Sample A is approved as either “extracorporeal membrane oxygenator for open heart surgery” or “ECMO for assisting circulation/ECMO for assisting respiration” in Japan. The requirement for the “ECMO for assisting circulation/ECMO for assisting respiration” is that “the membrane characteristics of the silicone membrane or the special polyolefin membrane can prevent plasma leakage” [9].

Figure 1 shows that sample A is wrapped with a vinyl chloride sheet as a measure against SARS-CoV-2 scattering from the gas outlet port of ECMO during the treatment for the severely ill patient with COVID-19. Vapor bubbles enclosed in the inner surface of the sheet were observed as early as 12 h after receiving ECMO support in all 10 patients, and plasma leakages were not visible from the membrane oxygenator.

As with the disposal of protective clothing, the sheet was discarded so that the outer surface of the sheet was not exposed.

Capiox^®^ CX-FX15E (Terumo Co., Ltd., Tokyo, Japan, Sample B) has built-in hollow fiber membranes made from polypropylene [12] and is approved as the “extracorporeal membrane oxygenator for open heart surgery”.

### 2.2. Observation of Anisotropic Pores and Cross Sections of the Hollow Fiber Membranes Using Field Emission Scanning Electron Microscopy (FE-SEM)

A FE-SEM (JSM-7610F, Jeol Ltd., Tokyo, Japan) was used to observe the cross sections and inner or outer surfaces of the hollow fiber membranes [19,20] at an accelerating voltage of 1.0 kV, a working distance of 15.0 mm, and an emission current of 47.2 μA in observation fields of magnifications of 250, 1000, 7500 and 30,000. The measurement mode was the LEI (lower secondary electron image). Samples were not sputtered with Au or C because it was confirmed that the voids in the porous membrane were crushed due to the radiant heat during the sputtering of Au. After immersing a hollow fiber membrane in a 50% aqueous acetone solution for 2 days, a sample was placed in liquid nitrogen and was sandwiched between two tweezers, and a hollow fiber membrane wall was broken to prepare a membrane cross-sectional sample. 

On the other hand, a hollow fiber was cut into a length of 1 cm and approximately a half in the longitudinal direction using a razor to observe the inner lumen and outer surfaces [27]. Samples were not sputtered with Au.

Pore (void in membrane wall) areas were measured in an observation field of a magnification of 30,000 in size by the analysis of digital imagery utilizing ImageJ software (National Institute of Health, Bethesda, MD, USA). The equivalent pore diameter and porosity of ECMO membranes were calculated using the values of pore areas.

### 2.3. Validation of SARS-CoV-2 Permeability Using the Steric Exclusion Model and Hindered Diffusion Model

An attempt was made to quantify the SARS-CoV-2 permeability through the membrane using the steric exclusion and hindered diffusion models [19,21]. The equations on models are briefly described here. The fraction of the pore cross-sectional area through which the solute penetrates is given by Equation (1).
(1)$$K=\frac{\pi {\left(r-a\right)}^{2}}{\pi r^{2}}={\left(1-\frac{a}{r}\right)}^{2}$$
where *K* is known as the solute partition coefficient, *a* is the molecular radius, *r* is the pore radius. The solute diffusive coefficient ($D_{m}$) across a membrane pore relative to the bulk solute concentration is defined as follows: (2)$$D_{m}=D_{AB}\left(\frac{K\omega_{r}}{\tau }\right)$$
where the $D_{AB}$ is the diffusive coefficient in water calculated by the Stokes–Einstein equation, the tortuous nature of the pore is explained by using the tortuosity (τ) for the actual pore length ($\tau \times \bar{L}$), $\omega_{r}$ is the hindered diffusion which depends on the ratio of the solute radius to the pore radius (a/r).

The solute permeability ($P_{m}$) is expressed in terms of the membrane properties (surface porosity (ε) and thickness ($\bar{L}$) of the membrane) as: (3)$$P_{m}=\frac{\varepsilon \cdot D_{m}}{\bar{L}}$$

## 3. Results

### 3.1. FE-SEM Observations of the Tortuous Pore Structures of the ECMO Membranes

Figure 2a,b shows FE-SEM images of ECMO membranes (dry), (a) the surface of the inner lumen of the capillary membranes, (b) the outer surface of the capillary membranes, respectively. Sample A (1), sample B (2), and BIOCUBE^®^ (NIPRO Co., Ltd., Tokyo, Japan) [19] (3) are shown for a comparison.

In Figure 2(a1,a2,b2), unique three-dimensional tortuous pore structures were observed. Sample A is made from poly (4-methyl-2-pentene) which is formed by the microphase separation (thermally induced phase separation (TIPS)). The pore structure of sample A is completely different from that of BIOCUBE^®^ because BIOCUBE^®^ is made from poly (4-methyl-1-pentene) (PMP) [7,18] by the melt-extruded stretch process [18]. Since (b1) is coated with poly-2-methoxyethylacrylate (PMEA) [13], no pores are confirmed. Sample B is made from polypropylene by the microphase separation process. The pores on the inner surface of sample B are close to the straight pore structure. The pore structure of sample B is also completely different from those of sample A and other ECMO membranes made from polypropylene [19] formed by the melt-extruded stretch processes. The outer surfaces of both sample A and sample B are coated with PMEA, but there is a difference, as shown in Figure 2(b1,b2). From this, the pore structure of the ECMO membrane is created by designing various membrane-forming processes using appropriate materials.

Figure 3 is the FE-SEM images of the cross sections and inner and outer surfaces of sample A (dry). Magnifications are (a) ×250, (b) ×1000, (c) ×7500, and (d–f) ×30,000, respectively. The cross section of sample A is the gradient pore structure that gradually becomes denser from the inside to the outside of the membrane wall. The polymer chain of the network structure made from poly (4-methyl-2-pentene) is thinner and smaller toward the outside of the membrane wall. Since the innermost layer of the cross section (d6) and the inner surface of the hollow fiber lumen (e) have similar structures, the three-dimensional pore structure near the surface of the hollow fiber lumen is observed almost accurately. On the other hand, although detailed observation of the outermost layer of the cross section is not sufficient, the outside of the cross section (d1) is different from the outer surface (f) of the hollow fiber membrane. The PMEA layer (f) is observed on the outer surface. Oxygen, carbon dioxide, and water vapor permeate through the outermost surface layer, however, SARS-CoV-2 does not permeate through the layer. They are all permeable through the voids of the gradient structure.

Figure 4 is the FE-SEM images of the cross sections and inner and outer surfaces of sample B (dry). Magnifications are (a) ×250, (b) ×1000, (c) ×7500, and (d–f) ×30,000, respectively. The cross section of sample B also has a gradient pore structure that gradually becomes denser from the inside to the outside of the membrane wall, but the change is more lenient than sample A. Unlike sample A, the innermost pore structure of the cross section of sample B (c4,d4) and the inner surface of the hollow fiber lumen (e) are completely different. In addition, the pore structure on the outer side (d1) of the cross section of sample B and the outer surface (f) of the hollow fiber membrane is completely different. The outer surface of the membrane (f) is coated with PMEA, but the coverage may be low (porosity may be high) due to insufficient coating of the PMEA on the outer surface.

As described above, following our previous research [19], the unique pore structures of the Capiox^®^ membranes are clarified. Elucidation of the pore structures of the ECMO membranes gives insights into the evolution of a next-generation ECMO membrane.

### 3.2. Determination of Pore Diameter, Surface Porosity, and SARS-CoV-2 Permeability

For a pore diameter determination, void areas that could be measured in observation fields of a magnification of 30,000 in size were analyzed (Figure 5). The areas of all the voids in each image were measured, and the membrane surface porosity of each image was calculated. In addition, an equivalent pore diameter was calculated for each void as a pore (Table 2). Furthermore, using the above technical data, the partition coefficient (K), the intramembrane diffusivity (D_m_), and the solute permeability (P_m_) of SARS-CoV-2 were calculated based on the steric exclusion model and the hindered diffusion model in the previous studies [19,21]. Figure 6 also shows the equivalent pore diameter distributions of samples A and B, and also shows the pore diameter (major axis) of BIOCUBE^®^ in the previous study [19] for comparison.

From Table 2 and Figure 6, sample A had a slightly wider distribution of equivalent pore diameter than that of sample B, and the average equivalent pore diameter of sample A was also larger than that of sample B. The surface porosity on the inner surface of the membrane was slightly smaller in sample A than in sample B, but no significant difference was observed (unpaired *t*-test). Using the same polymethylpentene as a material, sample A, which was formed by a microphase separation process, had a larger pore diameter than BIOCUBE^®^, which was formed by a melt-extruded stretch process. The equivalent pore diameter distribution on the outer surface of sample B was narrower than that on the inner surface, and the average equivalent pore diameter on the outer surface was also smaller.

The problem is, that, since the pore structures of the ECMO membranes in Figure 2 are very unique and complicated, the voids are difficult to be characterized as pores in Figure 5. Moreover, since it is difficult to measure all the voids’ areas in an image, the porosity may be underestimated. Although it may not be sufficient for quantitative evaluation in this approach, verifying the unique pore structure and membrane-forming process of the ECMO membranes will lead to the membrane-forming principle.

The diameter of SARS-CoV-2 is 50–200 nm [10]. Table 2 shows the partition coefficient, the intramembrane diffusivity, and the permeability calculated with the diameter of SARS-CoV-2 at 50 nm. Since no pores are observed on the outer surface of sample A, these values cannot be calculated. The partition coefficient of sample B is 0.49, thus SARS-CoV-2 permeates through the membrane wall when plasma leakage, plasma wetting (permeation of plasma into the membrane wall), and vapor permeation (permeation or penetration of water vapor into the membrane wall) occur. The intramembrane diffusivity of SARS-CoV-2 in plasma was 2.0 × 10^−12^ m^2^/s. The oxygen diffusivity in water is 2.76 × 10^−9^ m^2^/s, and glucose diffusivity in water is 9.30 × 10^−10^ m^2^/s (the values in the literature). The values calculated based on the model [19,21] are 3.26 × 10^−9^ m^2^/s and 4.68 × 10^−10^ m^2^/s respectively, so the credibility of the values in Table 2 is reasonable.

## 4. Discussion

### 4.1. Risk of SARS-CoV-2 Scattering due to Vapor Permeation in Capiox^®^ LX2

In Figure 1, sample A is covered with a sheet to prevent SARS-CoV-2 scattering, and it is confirmed that vapor condensation occurs on the inner surface of the sheet. Room temperature gas (relative humidity 0%) flows into the membrane oxygenator and is warmed by contact with blood through the membrane wall, and at the same time, the relative humidity rises to 100% due to the water vapor that permeates from the blood side through the membrane wall. When this gas flows out of the membrane oxygenator, it is cooled to room temperature and vapor condensation occurs in the gas side (outlet). Condensed water droplets or water vapor were scattered from the outlet port of the gas side, causing vapor condensation on the inner surface of the sheet in Figure 1. Therefore, it was inferred that water vapor permeated through the PMEA coating layer on the outermost surface (Figure 2(b1) and Figure 3f) of sample A. This phenomenon is explained by the solution–diffusion model [23], similar to the permeation of oxygen and carbon dioxide through the outermost surface coating layer.

In this case, if the water vapor contains SARS-CoV-2, there is a risk that it diffuses as an aerosol to cause a secondary infection. Medical professionals involved in ECMO treatment have been concerned about this secondary infection risk (ECMO device infection caused by ECMO, ECMO infection). However, since there are no pores on the outer surface of sample A that allow SARS-CoV-2 to penetrate the membrane wall, there is little risk of SARS-CoV-2 permeation through the membrane wall. In the cases where SARS-CoV-2 positive reactions were detected from the gas outlet port, even when plasma leakages were not visible from the membrane oxygenator [4], these were likely due to the SARS-CoV-2 permeation through the membrane wall associated with vapor permeation. On the other hand, sample B has pores that allow SARS-CoV-2 to penetrate the membrane wall, but sample B is for open heart surgery (Table 1) and is not used for ECMO treatment in severely ill COVID-19 patients. These insights into gradient pore structures of Capiox^®^ gas exchange membranes and SARS-CoV-2 permeation were revealed in this study, inspired by our previous works [19,27].

The risk of SARS-CoV-2 permeation through the ECMO membrane has been focused on during the COVID-19 pandemic. Furthermore, in this study, we focused on the risk of secondary infection of SARS-CoV-2 by water vapor permeation.

Since condensation of water vapor (wet lung) and plasma leakage occur due to water vapor that permeates through the membrane wall from blood to the gas flow path in the first place, it is required that water vapor does not permeate through the membrane wall in order to improve the durability of the membrane. However, at present, because the durability of the membrane during continuous long-term use is insufficient, especially in a VV-ECMO for acute respiratory failure, the theoretical advantage of sample A could decrease day by day. Future research should focus on the challenges to rule out the risk of SARS-CoV-2 contamination in the environment after one or two weeks of ECMO respiratory support. It is also necessary to put into practical use, the next-generation ECMO and ECMO devices that have durability and that do not cause such problems, even during long-term support exceeding 6 h. There has been a report of a clinical trial of the ultra-compact durable ECMO system for 2 weeks [31].

The durability impaired by the occurrence of water vapor permeation and plasma leakage is affected not only by the membrane structure but also by the flowing blood and gas. Therefore, it is also desirable to explain how the blood is flowing outside the hollow fiber and to analyze the relationship between the flowing blood and gas and the module design of Capiox^®^ [30].

Thrombosis is also a problem in ECMO support. Thrombosis may be evaluated by destroying the device after use and observing the blood components to the membrane [32].

### 4.2. Usefulness and Limitation of Theoretically Validating SARS-CoV-2 Permeability from the Perspective of Membrane Engineering

In this study, we attempted to quantify the SARS-CoV-2 permeability through the ECMO membrane wall using the steric exclusion model and the hindered diffusion model [19] for molecular permeation through membranes. These are classic models in membrane engineering, and simply verify the SARS-CoV-2 permeability through the ECMO membrane by comparing with useful literature values, such as oxygen or glucose diffusivity.

However, in this study, data, such as the type and physical characteristics, and affinity with the membrane and blood concentration of the SARS-CoV-2, could not be used. As soon as such data are accumulated, more detailed verification is needed. Ideally, a model experimental verification actually should be made to confirm SARS-CoV-2 permeability [33], however, it is not realistic to verify it at this time. On the other hand, in the medical field, strict measures are required day by day, and the above-mentioned data are required. Therefore, despite problems, the output of our membrane engineering approach will provide novel insights to the ECMO support of COVID-19 critically ill patients [1,2,3,4].

## 5. Conclusions

Both samples A and B have anisotropic pore structures which are completely different on the inner and outer surfaces of the hollow fiber membranes. The cross sections of samples A and B gradually become denser from the inside to the outside of the membrane wall; they are gradient pore structures. The pore structure of sample A is completely different from that of the PMP membrane formed by the melt-extruded stretch process. In sample A, water vapor permeates through the coating layer on the outer surface and the voids of the gradient structure. However, since there are no pores on the outer surface of the membrane that allows SARS-CoV-2 to penetrate, there is no concern that SARS-CoV-2 may permeate through the membrane wall during ECMO support.

## Figures and Tables

**Figure 1 membranes-12-00314-f001:**
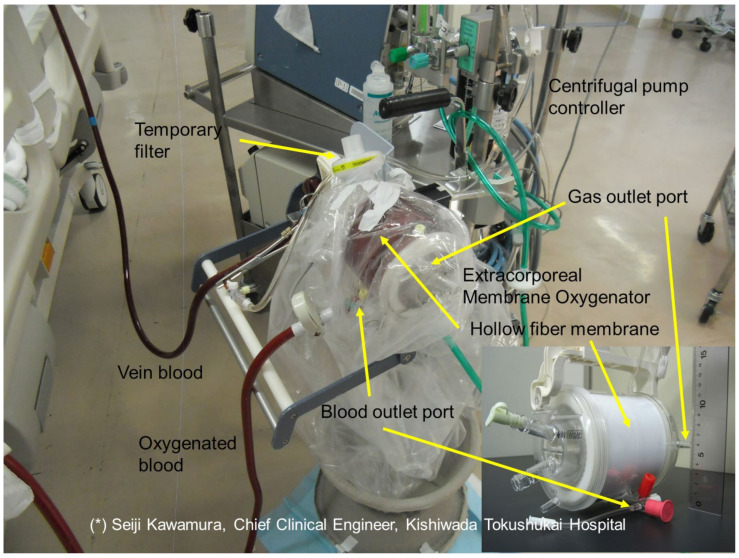
A measure to prevent the scattering of SARS-CoV-2 from extracorporeal membrane oxygenator, wrapped with a vinyl chloride sheet.

**Figure 2 membranes-12-00314-f002:**
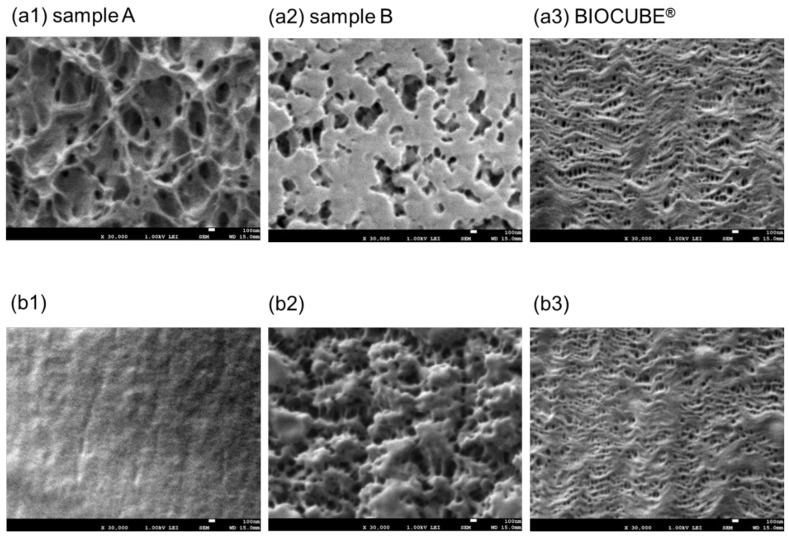
FE-SEM images of Capiox^®^ LX2 (dry), FX (dry), and NIPRO BIOCUBE^®^ (dry), (**a**) the surface of the inner lumen of the capillary membranes, (**b**) the outer surface of the capillary membranes, respectively, ×30,000. (1) sample A (Capiox^®^ LX2), (2) sample B (Capiox^®^ FX), and (3) BIOCUBE^®^.

**Figure 3 membranes-12-00314-f003:**
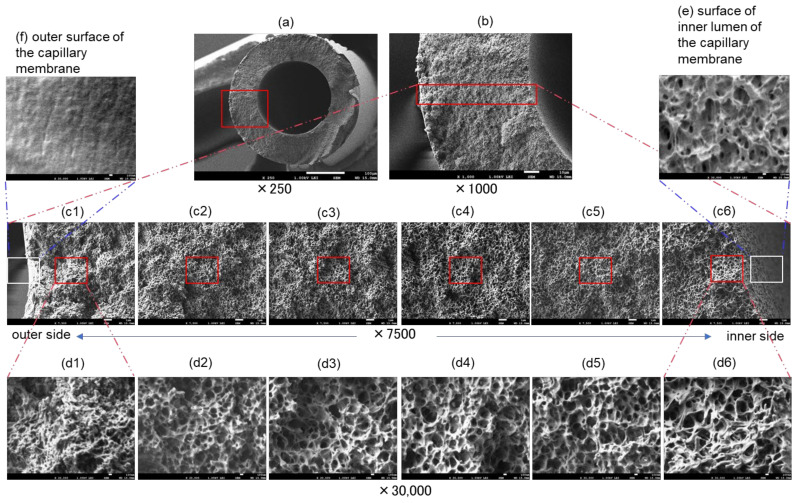
FE-SEM images of Capiox^®^ LX2(dry), (**a**–**d**) cross sections and (**e**) inner and (**f**) outer surfaces of the capillary membrane. Magnifications are (**a**) ×250, (**b**) ×1000, (**c**) ×7500 and (**d**–**f**) ×30,000, respectively.

**Figure 4 membranes-12-00314-f004:**
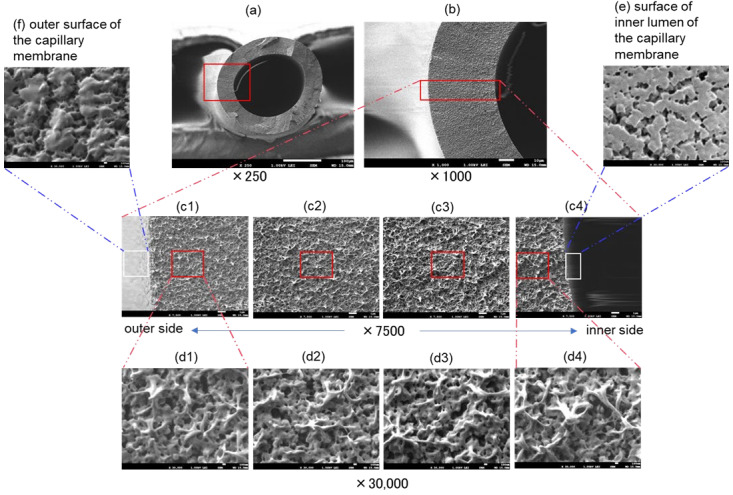
FE-SEM images of Capiox^®^ FX(dry), (**a**–**d**) cross sections and (**e**) inner and (**f**) outer surfaces of the capillary membrane. Magnifications are (**a**) ×250, (**b**) ×1000, (**c**) ×7500 and (**d**–**f**) ×30,000, respectively.

**Figure 5 membranes-12-00314-f005:**
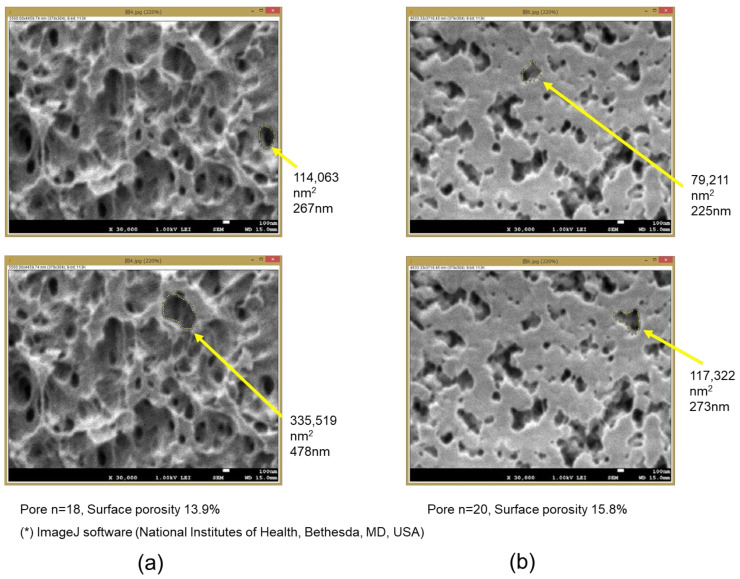
Measurement of pore areas and calculation of equivalent pore diameter and membrane surface porosity. (**a**) The inner lumen surface of sample A, (**b**) the inner lumen surface of sample B, respectively.

**Figure 6 membranes-12-00314-f006:**
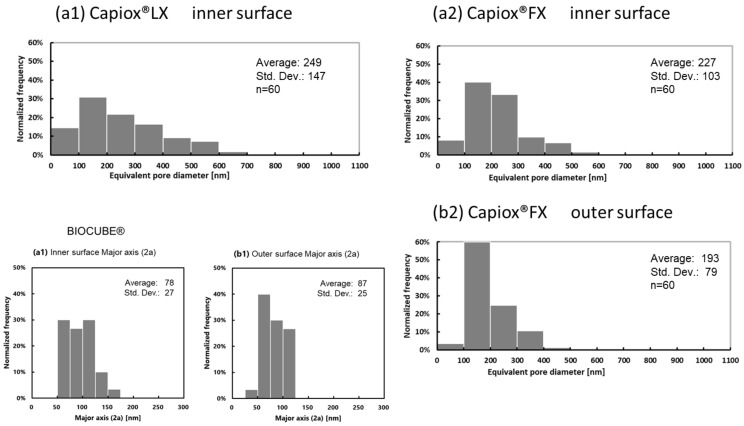
Distribution of the equivalent pore diameter of ECMO membranes (dry), determined via FE-SEM, (**a**) the surface of the inner lumen of the capillary membranes, (**b**) the outer surface of the capillary membranes, respectively. (**1**) Capiox^®^ LX, (2) Capiox^®^ FX and BIOCUBE^®^.

**Table 1 membranes-12-00314-t001:** Specification of hollow fiber membranes for membrane oxygenators.

Sample	Capiox^®^ CX-LX2 LW Sample A	Capiox^®^ CX-FX15E Sample B	BIOCUBE^®^ C 6000P [19]
Manufacturer (Manufacturer of membrane)	Terumo Co., Ltd., Tokyo, Japan	Terumo Co., Ltd., Tokyo, Japan	NIPRO Co., Ltd., Tokyo, Japan (DIC Co., Ltd., Tokyo, Japan)
Material of hollow fiber membrane	Poly(4-methyl-2-pentene) (PMP)	Polypropylene (PP)	Poly(4-methyl-1-pentene) (PMP)
Antithrombogenic material coating for blood flow path	Poly-2-methoxyethylacrylate (PMEA)	Poly-2-methoxyethylacrylate (PMEA)	Heparin
Inner diameter of lumen [µm] (*n* = 30)	207 ± 8	181 ± 10	176 ± 6
Membrane thickness [µm] (*n* = 30)	87 ± 3	50 ± 6	30 ± 2
Pore structure	skinned asymmetric pore structure	asymmetric pore structure	asymmetric pore structure
Sterilization method	EOG	EOG	EOG
Insurance coverage classification (in Japan)	extracorporeal membrane oxygenator for open heart surgery; ECMO ^(1)^ ECMO for assisting circulation/ECMO for assisting respiration ^(2)^	extracorporeal membrane oxygenator for open heart surgery; ECMO ^(1)^	extracorporeal membrane oxygenator for open heart surgery; ECMO ^(1)^ ECMO for assisting circulation/ECMO for assisting respiration ^(2)^

^(1)^ Usage time 6 h. ^(2)^ Usage time 6 h, the membrane characteristics of a silicone membrane or a special polyolefin membrane can prevent plasma leakage.

**Table 2 membranes-12-00314-t002:** Equivalent pore diameter of ECMO membranes and SARS-CoV-2 permeability.

Sample	Capiox^®^ CX-LX2 LW Sample A	Capiox^®^ CX-FX15E Sample B	BIOCUBE^®^ C 6000P
Equivalent pore diameter of inner surface [nm] *n* = 60, AVG ± STD.	249 ± 147	227 ± 103	78 ± 27 [19]
Equivalent pore diameter of outer surface [nm] *n* = 60, AVG ± STD.	NA	193 ± 79	87 ± 25 [19]
Porosity of inner surface [%] ^(1)^ *n* = 3, AVG ± STD.	12.7 ± 0.9	14.3 ± 1.2	18.2 ± 2.4
Porosity of outer surface [%] *n* = 3, AVG ± STD.	NA	13.8 ± 2.6	10.9 ± 1.0
Tortuousity [-] ^(2)^	1.0	1.0	1.0
Partition coefficient (K) of SARS-CoV-2 [-]	NA	0.49	0.12
Intramembrane diffusion coefficient (D_m_) of SARS-CoV-2 ^(3)^ [m^2^/s]	NA	2.0 × 10^−12^	8.9 × 10^−14^
Permeability (P_m_) of SARS-CoV-2 ^(4)^ [m/s]	NA	3.1 × 10^−9^	3.3 × 10^−10^

^(1)^ Unpaired *t*-test (*p* < 0.05, sample A versus sample B, inner surface versus outer surface of sample B). ^(2)^ Tortuousity: P_m_ was calculated with the tortuousity on the outer surface side of the membrane as 1. ^(3)^ Oxygen diffusivity in water: 2.76 × 10^−9^ m^2^/s, glucose diffusivity in water 9.30 × 10^−10^ m^2^/s (the value in the literature) [21]. ^(4)^ Calculated oxygen diffusivity in water using this model 3.26 × 10^−9^ m^2^/s (molecular diameter 0.2 nm), glucose diffusivity in water 4.68 × 10^−10^ m^2^/s (molecular diameter 0.7 nm). NA; not available.

## Data Availability

Not applicable.

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
