# Peer review of "Insights into Gradient and Anisotropic Pore Structures of Capiox® Gas Exchange Membranes for ECMO: Theoretically Verifying SARS-CoV-2 Permeability"

_membranes, 2022, doi:10.3390/membranes12030314_

Round 1

Reviewer 1 Report

This article deals with the insights into the Capiox® gas exchange membranes and their application in treating SARS-CoV-2. The article focuses on the characterisation of gas exchange membranes for ECMO and the risk of SARS-CoV-2 permeability through the membrane back to the environment. Overall, the article is very interesting, well structured and presents a thorough characterisation, however, there are some issues that need to be improved before publication. I suggest the publication after minor revision.

  1. Introduction
    • The introduction section is slightly too short. Especially, the last paragraph with the objective of the study. It should be broaden a little bit. Moreover, there are some grammatical errors throughout the entire document (repetitions, grammar etc.). The authors should carefully read the text once more and correct the errors.
  2. Materials and Methods
    • What were the outlet pipes (where the vein blood entered and oxygenated blood left) made of (Figure 1)? How do the authors ensure there was no escape of the virus through these pipes?
    • Section 2.1: It should be better explained how the blood is flowing inside the hollow fiber: Please use the membrane appropriate terminology, such as: lumen side, shell side etc. Other parameters should be added, such as: blood flow velocity, pressure drop, volume of the blood inside the device etc.
    • Section 2.3: The equations should be explained in a better way. It is not explained what is DAB, τ, ωr, K, a, π and others. I see there is a section with abbreviations at the end of the document but in general the importance of such equations should be mentioned and how exactly they were used.
  3. Results
    • How were the images a1, a2 and a3 obtained (Figure 2)? Were the hollow fibers cut across the length and “opened” to see the inner side? If so, that should be clearly explained in the experimental methods.
    • Figure 5: What are the membranes shown in this figure? A or B? Please, indicate clearly.
    • Table 2: Why are there so many missing information about Sample A?
    • Figure 6: Please make a uniform nomenclature. Sometimes the authors use Sample A and Sample B and sometimes Capiox LX or FX. It is confusing. Please choose one and unify across the entire document.

Reviewer 2 Report

The Manuscript (membranes- 1607520) entitled "Insights into Gradient and Anisotropic Pore Structures of  Capiox® Gas Exchange Membranes for ECMO; Theoretically Verifying SARS-CoV-2 Permeability" submitted to membranes investigated the permeability of SARS-Co V-2 through two types of the membranes . The study’s object is interesting but it needs to be revised before publishing. The following comments might be helpful to improve the quality of the manuscript.

1# It is mentioned that the both samples (sample A and sample B), are produced by the microphase separation processes using polymethylpentene (PMP) and polypropylene (PP), respectively. It would be better to clarify what phase separation method was applied for these membranes. The phase separation method includes non-solvent induced phase separation (NIPS) and thermally induced phase separation (TIPS).

2# Another membrane, which is made by NIPRO BIOCUBE, was mentioned in the paper but the specification of this membrane was not reported in Table 1 and Table 2 and…. It is highly recommended that the characterization of this membrane also report and compare with sample A and Sample B.

3# The thrombosis is a problem in the ECMO treatment. How did authors evaluate this problem for different samples (A & B)?

4# What is the procedure for disposing of the vinyl chloride sheet that wrapped the outlet ports of ECMO?

5# In page 2, “Vapor bubbles enclosed in the inner surface of the sheet were observed as early as 12 hours after receiving ECMO support in 10 patients, and plasma leakages were not visible from the membrane oxygenator.” How many patients were treated  in this study? What is the statistical method in this study?

6# What is the difference between the PMEA coating in sample A and B? Why is the PMEA coating of sample B not as good as sample A?

7# The sample A and B are suitable for membrane oxygenator application in comparison with other commercial membranes?

8# According to the results, what is the suggestion to decrease secondary infection risk in ECMO treatment?

Reviewer 3 Report

Dear Authors, I’ve read and appreciated your manuscript so much. Your research is very detailed and your conclusions are based on solid assumptions. 

The Figures 2-3-4 are very interesting: It's easy to understand the difference between the porosity of inner and outer surface of different ECMO membranes; so it's easy to agree with your conclusions.

Before editing I’ve only few request in following item: 

  • Line 201 : in Fig. 5 I don’t understand if the figure are related to Sample A and Sample B , or to Sample A and BIOCUBE. I wonder if you could better specify in your manuscript 

Finally in my opinion, it could be better to underline, that this theoretical Sample A advantage could decrease day by day, because of membrane’s continuous use, especially in a Venous-Venous ECMO for acute respiratory failure. 

So future researches could be published about this item, to disclose or to rule out the risk of Corona virus environment contamination’s risk after one or two week of ECMO respiratory support. 

Best Regards 

Round 2

Reviewer 2 Report

The authors have tried to revise the manuscript according to my comments.

The revised manuscript is accepted in the present form. However, before publishing it, the term of DAB in Eq. 2 should be described in the text.
